# Natural language processing for cognitive therapy: Extracting schemas from thought records

**Franziska Burger** [1] *, **Mark A. Neerincx** [1,2], **Willem-Paul Brinkman** [1]

1 Department of Intelligent Systems, Delft University of Technology, Delft, Netherlands, 2 Department of Perceptual and Cognitive Systems, Nederlandse Organisatie voor Toegepast Natuurwetenschappelijk Onderzoek (TNO), Soesterberg, Netherlands

* f.v.burger@tudelft.nl

**Data Availability Statement:** All data and analysis scripts are available from the 4TU.ResearchData repository (DOI:10.4121/16685347 / https://data.4tu.nl/articles/dataset/Dataset_and_Analyses_for_

## Abstract

The cognitive approach to psychotherapy aims to change patients' maladaptive schemas, that is, overly negative views on themselves, the world, or the future. To obtain awareness of these views, they record their thought processes in situations that caused pathogenic emotional responses. The schemas underlying such thought records have, thus far, been largely manually identified. Using recent advances in natural language processing, we take this one step further by automatically extracting schemas from thought records. To this end, we asked 320 healthy participants on Amazon Mechanical Turk to each complete five thought records consisting of several utterances reflecting cognitive processes. Agreement between two raters on manually scoring the utterances with respect to how much they reflect each schema was substantial (Cohen's $\kappa = 0.79$). Natural language processing software pretrained on all English Wikipedia articles from 2014 (GLoVE embeddings) was used to represent words and utterances, which were then mapped to schemas using k-nearest neighbors algorithms, support vector machines, and recurrent neural networks. For the more frequently occurring schemas, all algorithms were able to leverage linguistic patterns. For example, the scores assigned to the *Competence* schema by the algorithms correlated with the manually assigned scores with Spearman correlations ranging between 0.64 and 0.76. For six of the nine schemas, a set of recurrent neural networks trained separately for each of the schemas outperformed the other algorithms. We present our results here as a benchmark solution, since we conducted this research to explore the possibility of automatically processing qualitative mental health data and did not aim to achieve optimal performance with any of the explored models. The dataset of 1600 thought records comprising 5747 utterances is published together with this article for researchers and machine learning enthusiasts to improve upon our outcomes. Based on our promising results, we see further opportunities for using free-text input and subsequent natural language processing in other common therapeutic tools, such as ecological momentary assessments, automated case conceptualizations, and, more generally, as an alternative to mental health scales.

Extracting_Schemas_from_Thought_Records_using_Natural_Language_Processing/16685347).

**Funding:** This work is funded by the 4TU research centre for Humans & Technology. There is no grant number associated with this funding. All authors are part of the research center. URL: https://www.4tu.nl/ht/en/.

**Competing interests:** The authors have declared that no competing interests exist.

# Introduction

E-mental health—delivering therapeutic interventions via information and communication technology—is regarded as a promising means of overcoming many barriers to traditional psychotherapeutic care. Yet, in a review of more than 130 scientifically evaluated e-mental health systems for depression, it was found that the technological state of the art of these systems is limited: even in recently developed systems, technology is often only used as a platform for delivering information to the patient. When the patient is asked to provide open, unconstrained textual information to the system, this information is typically either processed by a human in the case of guided systems or not processed at all in the case of autonomous systems [1]. Although both methods are arguably very robust to misunderstanding, human processing is costly while no processing offers no advantage over traditional paper-based workbooks. However, developments in data-driven natural language understanding are increasingly able to reliably interpret unconstrained qualitative user input. Here, we explore this opportunity for a specific therapeutic task in cognitive therapy: determining underlying maladaptive schemas from the information contained in thought record forms.

Thought record forms provide patients with a structured format for monitoring their thoughts, consisting of descriptions of the thought eliciting situation, the experienced emotion, the first cognitive appraisal of the situation, and the resulting behavior. Thought records are commonly employed in cognitive therapy, a form of psychotherapy based on Beck's cognitive theory [2]. The theory posits that not the situations but the way in which we appraise them causes our emotions. For example, it is not the fact that we are not invited to a party that makes us upset but rather the fear or understanding that this says something about us or our relationship with the host. Our immediate and unreflected appraisal of a situation is called an automatic thought. Automatic thoughts are in turn determined by schemas, the cognitive structures that make up our world view. A specific schema can be activated given the right trigger. In people with certain mental illnesses, it is theorized that pathogenic schemas have a particularly low activation threshold [3]. Consequently, a core part of cognitive psychotherapy involves teaching patients to monitor thoughts for insight into underlying schemas. Starting from the automatic thought noted down in the thought record, the downward arrow technique (DAT) [4] helps to determine the causative maladaptive schema. It consists of repeatedly asking *why it would be upsetting* or *what would be the worst that could happen* if the idea stated in the previous step was true. An example thought record that we collected in our experiment is shown in Table 1. The DAT is illustrated by the final three rows (in cursive font). Since the majority of thought records in our dataset include the DAT, hereinafter the term *thought record* refers to both the core thought record and DAT unless explicitly stated. Also extending beyond the nomenclature typically used in clinical psychology, we define as a thought record *utterance* the automatic thought or any completed step of the DAT. Each of the final four rows of the Participant Response column of Table 1 reflects an utterance. As can be seen in the response to the second downward arrow step, i.e., *I want friends. I will be lonely otherwise.*, an utterance can consist of multiple sentences.

Unlike automatic thoughts, schemas have received little attention in empirical research to date [5]. When considered, they have typically been explored in a top-down manner with measurement instruments developed on the basis of cognitive theory and validated with exploratory factor analyses (for example, [5, 6]). To the best of our knowledge, only one classification rubric for schemas exists that was not exclusively derived from theory but created from a content analysis of a set of thought records (also including DAT) collected with an online self-help cognitive behavioral therapy (CBT) program, namely the schema rubric of Millings and Carnelley [7].

**Table 1. Example of one complete thought record from the dataset collected in this study.**

| TR Question | Entry Type | Participant Response |
|---|---|---|
| Describe the situation very briefly in your own words. | open text entry field | while walking down the street I see someone I know, wave at them and they don't acknowledge my wave. |
| How well can you imagine yourself in this situation? | slider from 0 (not at all) to 100 (as good as if you were in the situation at the very moment) | 85 |
| Describe your emotion in this situation in one word. | open text entry field | disappointment |
| How intensely would you be experiencing this emotion? | slider from 0 (a trace) to 100 (the most intense possible) | 45 |
| Which of the following four emotions corresponds best with the emotion that you wrote down above? | multiple choice: sadness, fear, anger, happiness | sadness |
| Which (automatic) thought might have caused you to feel this way in the described situation? | open text entry field | They don't like me enough to wave back |
| *And why would it be upsetting to you if "They don't like me enough to wave back" were true? What would it mean to you? What would it say about you?* | *open text entry field* | *I may be unlikeable.* |
| *And why would it be upsetting to you if "I may be unlikeable" were true? What would it mean to you? What would it say about you?* | *open text entry field* | *I want friends. I will be lonely otherwise.* |
| *And why would it be upsetting to you if "I want friends so I won't be lonely." were true? What would it mean to you? What would it say about you?* | *open text entry field* | *If I am unlikeable then I won't have friends and will be alone all my life.* |
| What would you do in the situation, if anything? | open text entry field | I would try to make better impressions on people I meet. |

Steps of the downward arrow technique are presented in cursive font. Three downward arrow steps were completed in this thought record. After each downward arrow step (question + open text entry field), participants were asked the intermediate closed question of whether they wanted to continue with the downward arrow technique or not. Thus, after each step participants indicated that they wanted to continue until the final one, where they indicated that they wanted to stop, thereby completing the downward arrow technique. The intermediate question is omitted here. Following the downward arrow technique, participants completed the entire thought record by describing their behavior in the situation. The scenario description presented to the participant was "You are walking down the street. On the other side of the street you see an acquaintance whom you've liked the few times you've been in his company. You wave to him, and you get no response."

In this work, we develop the natural language processing (NLP) foundation for a task-oriented conversational agent (CA) that motivates users to regularly complete thought recording homework exercises. Most CAs used in practice to date are frame-based [8, Ch. 24]. To be able to parse the semantics of a user input (e.g. "I want to take my girlfriend to the theater next weekend.") and fill the slots in a frame (e.g. day, show, theater, time, number of tickets), the agent needs to classify broadly the intent of the entire input phrase (e.g. book theater tickets) and extract specifically the information corresponding to empty slots. When all slots are filled, the agent can complete the task. Up until recently, intent classification and slot filling were mostly done using a hand-written, domain-specific semantic grammar, often prescribing possible synonyms as well as a certain order for the information (e.g. {I want | Could I | It would be great if I could} * {book | reserve | get} * {tickets | cards} * {movies | theater} *). Systems using such grammars are expensive in terms of engineering time and prone to errors and misunderstandings [8, Ch. 24]. Both drawbacks have been largely eliminated with the advent of deep learning in the past decade. Rather than hand-crafting large sets of rules, deep learning allows for the acquisition of synonyms and word usage in context from large sets of data, such as Wikipedia. As two recent literature reviews show, these developments are slowly finding their way into CAs for health care [9, 10]. Laranjo et al. [9] found most of the CAs allowing for unconstrained natural language input to have been developed after 2010. Yet, only one-half of

the reviewed agents used frame-based or agent-based dialog management [11], while the other half implemented entirely system-driven and finite-state dialog management strategies. The authors therefore conclude that CAs in health care are not up to par with those in other fields. Of all 40 agents considered in [10], only six use state-of-the-art natural language understanding techniques [12–17].

While it is always important to limit user frustrations that arise from understanding errors on the part of the CA, this is particularly crucial in dialog systems for mental health treatment due to the highly emotionally sensitive domain. It is conceivable that language understanding errors as well as inconsistent or insensitive [18] responses could affect not only patients' experience and trust in the system but, in the worst case, also their mental health. Consequently, rule-based systems have been the norm [19]. Questions from the system are phrased so narrowly that they leave little room for unexpected responses (e.g., [20]). Since even therapy following a strict protocol is much less task-oriented than booking theater tickets, most systems fully or partially resort to providing multiple response options to the user (see, for example, [21]). The more recently developed Woebot [22], a chatbot for treating college students with symptoms of anxiety and depression, only uses natural language processing as an option for some nodes of Woebot's decision tree architecture, choosing the next node mostly based on user selection of one of several suggested replies.

Thought recording exercises are often assigned as homework to patients in face-to-face treatment or included in self-help workbooks and treatment systems with only general instructions. Timely feedback or tailored support from a therapist are therefore usually not available when patients attempt the exercise. As the goal of thought recording is the discovery of thinking patterns, frequent completion of thought records is crucial for their success. It is for these reasons that we aim to build a CA to motivate and support people in regularly completing thought records. The CA can use knowledge about schemas to provide feedback, respond understandingly, or to strategically ask for supplementary information. This work therefore addresses the following primary research question: Can the underlying maladaptive schema of a thought record utterance be scored by a machine?

## Hypotheses

The objective of this study was to see whether identifying schemas from thought records is at all possible. Consequently, our first hypothesis is that schemas can be extracted automatically (H1). We investigate this with a future goal of implementing a conversational agent capable of providing useful feedback. For such practical applications, we were also interested in studying ways to potentially improve automatic schema identification. As a result, three additional hypotheses, informed by psychological theory, were also investigated: automatic predictions improve as the downward arrow technique progresses (H2), within individuals, similar situations will activate the same schemas (H3), across individuals, there is a relationship between the active schemas and scores on mental health scales (H4). We here motivate the hypotheses in turn.

### H1: Schemas can be extracted automatically

As outlined above, conversational agents in health care, and particularly in depression treatment, to date are employing grammar-based or no NLP more often than not. Yet, the field more generally has not been blind to state-of-the-art data-driven methods. Thus far, however, they are mostly used in clinical psychology research to perform psychological assessment. Social media platforms and forums provide a treasure trove of natural language data occurring in virtual social environments. This has resulted in a large body of literature searching for

linguistic markers indicative of depression, crisis, or suicidal risk in the data (e.g. [23–29]). One such example is the crisis detection models developed in [27]. With a dataset of posts comprising on average three sentences collected through the mental health app *Koko*, the authors use a recurrent neural network (RNN) to detect crisis (binary classification task). They augment their RNN with *attention* [30] to display the parts of a post that the neural network pays attention to during classification. Their best model, an RNN without attention, detects crisis with an F1-score accuracy of 0.80. In another study [28], the task was to correctly identify which topic-based forum (or *subreddit*) on the social media website Reddit the posts of users belong to. The posts were drawn from eleven different manually selected mental health sub-reddits. The best performing algorithm achieved an F1-score accuracy of 0.71 with a convolutional neural network in this multi-class (more than two classes that are mutually exclusive) classification task. Benton et al. [29] study a similar problem as a multi-label (more than two classes that are not mutually exclusive) learning task. Using tweets posted on the social media platform Twitter, they simultaneously classify suicidal risk, atypical mental health, and seven mental health conditions. They observed a clear added benefit of leveraging possible correlations between the labels in the multi-label models compared to a set of nine single-class prediction models. Although the described research indicates that automatically identifying crisis or mental health conditions from social media corpora is feasible, it is unknown whether this applies to schemas as well. However, the fact that the schema rubric of Millings and Carnelley [7] was obtained via content analysis from a corpus of thought records indicates that language and word usage differ between the schemas. If this is the case, a good model trained on sufficient data should be able to pick up on these differences. Additionally, schemas are not mutually exclusive and might therefore inform each other, possibly further improving prediction accuracy. On the basis of these considerations, we posit the following:

H1 The schema(s) underlying a thought record can be identified by an algorithm with an accuracy above chance.

## H2: Downward arrow converges and H3: Schema patterns are similar across thought record type

Thought records ask patients to first briefly describe the situation that resulted in the pathogenic emotion in their own words. The automatic thought is thus directly connected to the situation description and both are highly individual. Automatically analyzing such free-form open text without any further restrictions is an *open-domain* NLP task, similar to small-talk. For an artificial intelligence, this is notoriously difficult to deal with well as it requires a comprehensive world model of many topics. Such a model cannot feasibly be engineered by humans and, if it is at all possible, very large amounts of data would be required to construct it bottom-up. Models created in this manner are usually no longer transparent and may show unintended behavior (e.g., [31]).

From a clinical perspective, an alternative to open thought recording is to elicit schemas by means of imagined situations, using scripted situation vignettes as a basis for the thought records. Thought recording is typically assigned as homework for the patient in cognitive therapy, with the completed forms constituting an integral part of the face-to-face sessions. While leaving patients to their own devices provides them with freedom and ensures ecological validity, the various different steps of the thought recording method do not always come easy to patients [32]. When they struggle, therapists may guide the process by resorting to imagery or role-play so as to recreate the situation in the face-to-face session and evoke the automatic

thought again [33]. For initial practice [34] or for the controlled assessment of cognitive errors [35–37] and cognitive restructuring skills [38], therapists may additionally restrict patients by asking them to envision themselves in certain scripted ambiguous scenarios. From a technical perspective, such a scripted scenario can delimit the natural language domain. Taking the scenario into account in a schema identification model should thus produce more reliable results. Despite scenarios being viable from a clinical perspective and the safer option from a technical perspective, two aspects of cognitive therapy give rise to the possibility of open classification models for this specific NLP task: the *downward arrow technique* and the categorization of situations into *situation types*.

**Downward arrow technique.** The theory behind the downward arrow technique (DAT) posits that as one progresses along the downward arrow, a schema will be reached. While automatic thoughts are specific appraisals of situations, schemas are general: the same schema can cause a large variety of specific automatic thoughts. From this, it should follow that the thoughts delineated with the DAT become increasingly independent of the situation description. For the NLP, this means that the language in utterances should converge to language that is more characteristic of the schema. We therefore hypothesize as follows:

H2 Schema identification accuracy increases as one proceeds along the downward arrow.

**Categorization of situations.** Two situation types that are commonly distinguished in cognitive therapy are interpersonal situations and achievement-related situations (e.g., [39]). *Interpersonal* situations pertain to one's self-worth in relation to other people, while *achievement-related* situations are such where one might perform poorly and one's self-esteem is at risk. Hence, a schema identification model might generalize to any real-world situation as long as it takes into account whether the situation type is more interpersonal or more achievement-related. Consequently, the following hypothesis is tested:

H3 Within an individual, the schema patterns of scenario-based thought records can predict those of the real-life thought record when they match in situation type (interpersonal or achievement-related).

## H4: Mental illnesses have associated schemas

Lastly, cognitive theory argues for differences between depression and anxiety with regard to schemas. Depressed individuals are theorized to have overly negative views of the self, the world, and the future, while anxious individuals hold schemas related to personal danger [40]. However, Millings and Carnelley [7] found that only the presence of the schema related to power and being in control differs between those with depression and those with anxiety, with particularly the anxious participants in their online CBT program presenting with the schema. If each mental illness were to show specific associated schemas, though, mental health data could inform a prior distribution over schemas in terms of their likelihood. This might improve a machine learning model. Using the coding scheme of [7], we therefore pose the following exploratory hypothesis:

H4 The schema patterns of an individual combined across thought records can predict his or her depression, anxiety, and cognitive distortions as self-reported using standard psychological questionnaires.

## Methods

To test the hypotheses stated above, a dataset of completed thought records was needed. Copies of thought records from actual patients gathered through a therapeutic practice were not an option because we could not obtain access to such an existing corpus. We therefore chose to collect a new dataset of thought records through the online crowdsourcing platform Amazon Mechanical Turk. The Human Research Ethics Committee of Delft University of Technology granted ethical approval for the research (Letter of Approval number: 546).

### Design

The data collection process was designed as a cross-sectional observational study. This means that there were no independent variables manipulated and consequently no conditions.

### Materials

Three online platforms were used in the study: Amazon Mechanical Turk (MTurk) for recruitment, Qualtrics for data collection, and YouTube for hosting instructional videos on how to complete thought records. People who registered for the task on MTurk were redirected to Qualtrics. YouTube videos were embedded in Qualtrics.

The instructions for the thought recording task included psychoeducation on cognitive theory, a short description of the components of a thought record, and four video examples of how to complete the thought records using two scenarios and four fictional characters to emphasize that thought records are highly individual and that there are no incorrect answers as long as thought records are coherent.

Two types of thought records were used in the study: *closed* and *open* thought records. The closed thought records asked participants to imagine themselves in a certain pre-scripted scenario and to write thought records as if what is detailed in the scenario had happened to them. The open thought records, on the other hand, asked participants to write thought records using a recent situation from their own lives. The scenarios of the closed thought records for any participant were chosen from a set of ten possible scenarios. These were divided into two sets of five scenarios, one set comprising scenarios of an interpersonal nature, the other comprising scenarios of an achievement-related nature. The scenarios were taken from the Ways of Responding Questionnaire [38] and the Cognitive Error Questionnaire [36]. A complete list of the scenarios can be found in the data repository of this study (DOI: 10.4121/16685347). The open thought record followed the exact same structure as the closed ones, except that participants had to briefly describe a situation that happened in their life instead of first imagining themselves in a given scenario and then describing it again in their own words.

The formulation of the downward arrow technique (DAT) questions depended on the emotion category that participants selected. When this was *happiness*, they were not directed to complete the DAT after stating the automatic thought. Therefore, all thought records in our dataset have at least one utterance: the automatic thought. When selecting *sadness* or *anger* the DAT consisted of repeatedly asking "And why would it be upsetting to you if [previously stated thought] were true? What would it mean to you? What does it say about you?" When selecting *fear*, on the other hand, the corresponding question was "And what would be the worst that could happen if [previously stated thought] were true? What would it mean to you? What does it say about you?" Just like the thought records, the DAT was altered slightly to better fit online administration: after each step, participants were asked whether they wanted to continue with the technique or not. This was necessary to eventually break the loop while giving participants the chance to complete as many steps as they wanted.

## Measures

Three mental health questionnaires were used: the Hospital Anxiety and Depression Scale (HDAS) [41], the Beck Depression Inventory (BDI-IA) [42], and the Cognitive Distortions Scale (CDS) [39]. The HDAS is a diagnostic tool for depression and anxiety, while the BDI-IA only assesses symptoms of depression. The CDS measures to what degree someone suffers from cognitive distortions, such as black-and-white thinking, in achievement-related as well as in interpersonal situations.

The post-questionnaire comprised three items asking participants how difficult and how enjoyable they found it to complete a thought record, and to indicate how many thought records they think they would complete if they were asked to complete a thought record daily for a period of seven days. We collected this data as secondary measures in anticipation of follow-up research, in which we aim to implement a conversational agent to motivate users to regularly record their thoughts. The data from the post-questionnaire were collected for follow-up research and will not be discussed in this paper.

## Participants

The only qualifications participants needed to access the task on MTurk was to be located in the USA, Canada, the UK, or Australia, to be at least 18 years of age, and to never have participated in the same study before. A total of 536 participants accepted the task on MTurk. Of these, 320 responses were usable. Hence, approximately 40% of responses had to be excluded on the basis of participants failing at least one of the two instruction comprehension questions or not taking the task seriously (having filled in incomprehensible text or obviously having copied and pasted text from other websites into the text-entry fields). Excluded participants were not reimbursed. Participants who completed the experiment received $4 for their participation, based on an estimated 35 minutes needed to read the instructions and to complete the task and all questionnaires. This estimate was obtained from a pilot run with 10 participants. In choosing the reimbursement amount, we aimed to fairly reimburse participants' time. As a consequence, the Amazon Mechanical Turk workers, just like patients wishing to get healthier, had an incentive to do the task. However, we did not use the reimbursement to motivate our participants to put in extra effort, as all participants received the same reimbursement.

Of the 320 included participants, 148 were female, 171 were male, and 1 indicated *Other*. The mean age of 319 participants was 36.25 years (SD = 10.99) with the youngest being 19 and the oldest 71. Demographic questions were optional and one participant chose not to provide her age.

## Procedure

Participants fulfilling the qualification criteria could access the task in MTurk. There, they were presented with basic information about the study, such as a short description of the task and the expected time to complete it. Once having accepted the task, participants were redirect to Qualtrics for the experiment. Upon giving their explicit consent to six statements, they were forwarded to a short demographic pre-questionnaire followed by the task instructions. To ensure that participants would not rush through the instructions, two instruction comprehension questions completed the instructional part: one asking participants what they would have to do in the main task in general and the other concerning procedural aspects of how to complete the thought records as explained in the videos. Failing to answer at least one of the questions correctly resulted in the immediate exclusion of the participant. This was made clear to participants before reaching the questions and the questions were displayed on the same page as the instructions, allowing participants to re-read instructions or re-watch videos before

giving their answer. Participants who answered both instruction comprehension questions correctly were forwarded to the thought recording task. This consisted of four closed and one open thought record in this order. For the closed thought records, they were asked to first read the short scenario description and imagine themselves in the situation. They were then directed to a new page with the first thought record form. Throughout the process of completing this, it was possible at any point for the participants to access a short version of the instructions again.

The thought record form was followed by the downward arrow technique. After each step of the DAT, participants were asked whether they wanted to continue with another step. This allowed repeatedly reminding them of the stopping criteria: repeating oneself or feeling that answers were becoming somewhat ridiculous. After indicating that they did not want to continue with the DAT or in case of having selected *happiness* as the emotional response to the situation, participants were presented with the final thought record question. This concerned the behavior they would expect themselves to exhibit in the situation. The post-questionnaire and the three mental health scales completed participation. The entire experimental flow is visualized in S1 Appendix.

## Data and analysis strategies

To obtain a labeled dataset for training the schema identification models, the thought record utterances had to be scored manually. To this end, we used the schema rubric developed by Millings and Carnelley [7]. This rubric comprises ten categories, of which nine are well-defined schemas, such as *Attachment* or *Meta-Cognition*. The final category, however, is an "other" category for all thought records that cannot be assigned one of the well-defined schemas. Schemas are not mutually exclusive, a thought record can therefore be labeled with multiple schemas. We made three modifications to the original rubric. The first modification pertains to the area of application: the original rubric is always applied to an entire thought record, while we apply it to thought record utterances. As a second modification, we dropped the *Other* category, but allowed utterances to have a 0-score for all of the nine schemas labels. As the final modification, we have altered the original rubric from an utterance being indicative of an underlying schema or not (binary schema label) to it being indicative of an underlying schema to a certain degree (ordinal schema score). The schema scores that we assign range from *has absolutely nothing to do with the schema* (0) over *corresponds a little bit with the schema* (1) and *corresponds largely with the schema* (2) to *corresponds completely with the schema* (3).

The schemas of thought record utterances and the scenario type of the open thought record had to be manually scored. Table 2 shows example thought record utterances from our dataset for each of the nine schemas and the nine scores assigned to each of the utterances. All manual scoring was conducted by the first author, who scored the utterances in random order. To obtain an indication of reliability, an additional coder, a graduate student of clinical psychology, scored a subset of the utterances. For this, three subsets of 50 randomly selected utterances were used to train the coder until agreement on the interpretation of definitions was reached. Any scoring deviation of more than one point on the ordinal scale was discussed. Then the second coder coded another subset of 100 randomly chosen utterances. Interrater agreement between the first and second coder on this subset was substantial (weighted Cohen's $\kappa$ = 0.79). The first coder also recoded the same subset one year after completing the initial coding of all utterances with good intracoder agreement (weighted Cohen's $\kappa$ = 0.83).

**H1: Schemas can be automatically extracted.**    To test the first hypothesis, thought record utterances were studied taking a natural language processing perspective: using a machine

**Table 2. Example utterances for each schema taken from the dataset collected in this study.** Utterances are thoughts and can be either automatic thoughts or any thought written as part of the downward arrow technique. Scores were manually assigned for each of the nine mental health schemas by the first author.

| Utterance | S1 | S2 | S3 | S4 | S5 | S6 | S7 | S8 | S9 |
|---|---|---|---|---|---|---|---|---|---|
| S1: Attachment examples | | | | | | | | | |
| I am unlovable and less than other people. I will never find friends or a girlfriend. | 3 | 0 | 3 | 0 | 0 | 0 | 0 | 1 | 0 |
| I don't want to be alone. | 3 | 0 | 0 | 0 | 0 | 0 | 0 | 0 | 0 |
| I was a bad mom. | 3 | 0 | 0 | 0 | 0 | 0 | 0 | 0 | 0 |
| I failed at the relationship. | 3 | 0 | 0 | 0 | 0 | 0 | 0 | 0 | 0 |
| I won't be a good partner to others. | 3 | 0 | 0 | 0 | 0 | 0 | 0 | 0 | 0 |
| S2: Competence examples | | | | | | | | | |
| I feel like a failure at my job. | 0 | 3 | 0 | 0 | 0 | 0 | 0 | 0 | 0 |
| I'm unprepared for this task. | 0 | 3 | 0 | 0 | 0 | 0 | 0 | 0 | 0 |
| I can never go into a sales job. | 0 | 3 | 0 | 0 | 0 | 0 | 0 | 0 | 0 |
| I am not good enough to get a job. | 0 | 3 | 0 | 0 | 0 | 0 | 0 | 0 | 0 |
| I would be unable to produce saleable work. | 0 | 3 | 0 | 0 | 0 | 0 | 0 | 0 | 0 |
| S3: Global self-evaluation examples | | | | | | | | | |
| It would mean that I am lazy and I need to do better | 0 | 0 | 3 | 0 | 0 | 0 | 0 | 0 | 0 |
| I should never have been born. | 0 | 0 | 3 | 0 | 0 | 0 | 0 | 3 | 0 |
| I am selfish. | 0 | 0 | 3 | 0 | 0 | 0 | 0 | 0 | 0 |
| S4: Health examples | | | | | | | | | |
| I would become ill. | 0 | 0 | 0 | 3 | 0 | 0 | 0 | 0 | 0 |
| I feel exhausted and anxious. | 0 | 0 | 0 | 2 | 1 | 0 | 0 | 1 | 0 |
| I cannot lose weight no matter what I try. | 0 | 0 | 0 | 3 | 2 | 0 | 0 | 0 | 0 |
| It would be very depressing, it would say that I would need counseling to get through life. | 0 | 0 | 0 | 3 | 1 | 0 | 0 | 1 | 0 |
| I will have health issues | 0 | 0 | 0 | 3 | 0 | 0 | 0 | 0 | 0 |
| S5: Power and control examples | | | | | | | | | |
| I'm going to be stuck in my current situation forever. | 0 | 0 | 0 | 0 | 3 | 0 | 0 | 1 | 0 |
| The feeling of being pressured by my boss. | 0 | 0 | 0 | 0 | 3 | 0 | 0 | 0 | 0 |
| I was fired and not given a chance to succeed. | 0 | 1 | 0 | 0 | 2 | 0 | 0 | 0 | 0 |
| I am not in control of what I do or how I perceive myself | 0 | 0 | 0 | 0 | 3 | 2 | 0 | 0 | 0 |
| That I still have a target painted on my back for their abuse. | 1 | 0 | 0 | 0 | 3 | 0 | 1 | 0 | 0 |
| S6: Meta-Cognition examples | | | | | | | | | |
| My perception of people is off and that's why I have a difficulty creating new relationships. | 1 | 0 | 0 | 0 | 0 | 3 | 1 | 0 | 0 |
| That I can be more than a bit compulsive about investigating odd byways of thought. | 0 | 0 | 0 | 0 | 0 | 3 | 0 | 0 | 0 |
| I trick myself into believing I'm better than I am. | 0 | 0 | 0 | 0 | 0 | 3 | 0 | 0 | 0 |
| Because I hold myself to a high standard. | 0 | 0 | 0 | 0 | 0 | 2 | 0 | 0 | 0 |
| I get angry easily over small things. | 0 | 0 | 0 | 0 | 3 | 1 | 0 | 0 | 0 |
| S7: Other people examples | | | | | | | | | |
| People would rather avoid me than be in my presence. | 0 | 0 | 0 | 0 | 0 | 0 | 2 | 0 | 3 |
| It means that these people not care about anyone but themselves, and i have to suffer | 0 | 0 | 0 | 0 | 0 | 0 | 3 | 0 | 0 |
| People will mock me | 0 | 0 | 0 | 0 | 0 | 0 | 3 | 0 | 3 |
| I am not as selfish as other people. | 0 | 0 | 0 | 0 | 0 | 0 | 3 | 0 | 0 |
| It means that other people can do despicable things and not be accountable. | 0 | 0 | 0 | 0 | 0 | 0 | 3 | 0 | 0 |
| S8: Hopelessness examples | | | | | | | | | |
| I will stop trying in life and give up | 0 | 0 | 0 | 0 | 1 | 0 | 0 | 3 | 0 |
| I should never have been born. | 0 | 0 | 3 | 0 | 0 | 0 | 0 | 3 | 0 |
| Depression makes me think I'd be better off dead. | 0 | 0 | 0 | 2 | 0 | 0 | 0 | 3 | 0 |
| I will never have a life I enjoy | 0 | 0 | 0 | 0 | 0 | 0 | 0 | 3 | 0 |
| I'll never feel like I have a purpose. | 0 | 0 | 0 | 0 | 0 | 0 | 0 | 3 | 0 |

*(Continued)*

**Table 2.** (*Continued*)

| Utterance | S1 | S2 | S3 | S4 | S5 | S6 | S7 | S8 | S9 |
|---|---|---|---|---|---|---|---|---|---|
| S9: Others views about self examples | | | | | | | | | |
| My friends don't like me. | 2 | 0 | 0 | 0 | 0 | 0 | 0 | 0 | 3 |
| Because I want people like him to like me. | 0 | 0 | 0 | 0 | 0 | 0 | 0 | 0 | 3 |
| I could not make him see that I am a responsible person. | 0 | 0 | 0 | 0 | 0 | 0 | 0 | 0 | 3 |
| I must not be his type of person. | 0 | 0 | 0 | 0 | 0 | 0 | 0 | 0 | 2 |
| It would say that she did not feel like she was able to talk to me. | 0 | 0 | 0 | 0 | 0 | 0 | 0 | 0 | 3 |

learning model to score an utterance with regard to the nine well-defined schemas. This task can formally be described as an ordinal multi-label scoring task: an algorithm must assign each utterance a schema vector consisting of nine values ranging between 0 and 3. Assigning ordinal scores to data is generally not trivial and common simplifications are to either treat the ordinal scores as separate classes (nominal data) or as equidistant integers on a continuum (interval data) [43]. The former is otherwise known as classification and entails that the ordering information of scores is lost. The latter is regression and entails that the ordering is maintained, but information is added, such as that labels are equally spaced and that the space between labels can be meaningfully interpreted. Where specific algorithms have been created for ordinal data [43], these often assume that higher ordinal labels subsume lower ones (compare, for example, [44]), e.g., if something corresponds very much to a schema (score 3) it also automatically corresponds a little bit to the schema (score 1). This is not the case here, as we also have score 0 meaning that an utterance does not correspond to a schema. Another criterion for choosing algorithms was the ready availability of functional, well-maintained, and commonly used software packages. We assume this to work to the advantage of reproducibility and further development. As a result of these considerations, we opted to explore both approaches of treating the scores as nominal as well as treating them as interval rather than exploring specific ordinal methods.

Before automatically scoring, the data were linguistically preprocessed by lower-casing, replacing misspellings, contractions, and numbers, adding missing sentence end marks and comma space, and finally removing stop words and unnecessary white space. They were then divided into a training set, a validation set, and a test set, with the test set comprising 15% of all data, the validation set comprising another 12.75%, and the training set comprising the remaining 72.25%. Samples to include in test and validation set were not selected at random but rather we ensured that three criteria were fulfilled: 1. similar distribution of schemas, 2. approximately the same proportion of open and closed scenarios, 3. approximately the same distribution over DAT depths as in the entire dataset. This was achieved by randomly sampling 1000 times from the entire distribution, determining the deviation in distribution between the sample and the population for each of the three criteria, summing these three deviation measures, and choosing the sample with the smallest result. The process was first done for the test set and then repeated with the remaining data samples to obtain the validation set. We used normalized, 100-dimensional GLoVE embeddings [45] trained on all English Wikipedia articles existent in 2014 to represent the words in utterances.

Three types of algorithms of varying levels of complexity were chosen for the task: k nearest neighbors classification (kNN-C) and regression (kNN-R), support vector machine classification (SVC) and regression (SVR), and a multi-label recurrent neural net (RNN) as well as a set of separate RNNs per schema. All three types of algorithms are supervised-learning algorithms, meaning that they learn from labeled examples. The k-nearest neighbors algorithms work as

follows: for each new utterance that the algorithm has to label, a distance is calculated between this utterance and each of the utterances of the training set. The distance indicates how similar, i.e. *close* in representation space, the new utterance is to the utterances the algorithm has seen before. In our case, the distance metric was calculated by first linguistically preprocessing each utterance, then representing each word of an utterance as a GLoVE-embedded word-vector, normalizing the vectors, averaging all word-vectors of an utterance, and finally computing the cosine similarity between the utterance and each utterances of the training set. The $k$ then determines the number of closest training utterances (neighbors) that will be taken into account when calculating the label for the new utterance, i.e., if $k = 5$, the five closest training utterances will be considered. In the case of kNN-C, we combine the scores of the k neighbors with a conservative *mode* function, i.e., the unseen utterance is assigned the score that the majority of neighbors carry and the one with the lowest value if multiple exist. In the case of kNN-R, we combine the values by averaging the scores of the nearest neighbors. The kNN algorithms serve as a baseline as they are not trainable, i.e., for each new utterance all distances to all training examples must be computed again and thus all training data must be stored.

The second set of algorithms we applied to the data are support vector machines (SVMs). Unlike kNN algorithms, SVMs build a model from the training data, after which the data can be discarded. They are particularly suited for high-dimensional feature spaces. The core idea of SVMs for classification lies in finding a linear separation boundary between classes such that the space between the closest training examples on either side of the decision boundary (the margin) is maximized. To this end, they can leverage kernel functions to map classes that are not linearly separable in a lower-dimensional space to a higher-dimensional space. In SVMs for regression, on the other hand, a regression is fit to the data. The aim to maximize the margin around the regression line such that the error remains below a certain threshold. For the SVM algorithms, we again represented the utterances as averages of word-vectors. These were then standardized and used to train separate SVMs for each schema.

The final set of algorithms we used to model the data were recurrent neural networks (RNNs). Neural networks commonly consist of an input and an output layer and any number of hidden layers. The input layer holds nodes that simply pass on the numerical representation of the data. Each further layer is comprised of nodes and connections that transform the input. Each node combines all the signals coming in from the previous layer (transfer function) and decide whether or not to pass a signal on (activation function). Nodes of one layer are connected with the nodes of the next layer via weighting functions that amplify or discount the signal by means of multiplication. The output layer holds nodes that transform the signal to the desired type of output value, e.g., a value between 0 and 1. Neural networks become deeper with each additional hidden layer. While feedforward networks, in which the signal travels only in one direction from input to output layer, cannot deal with sequential input data, RNNs are a type of deep neural network specifically designed for this purpose. Thus, unlike the kNN and the SVM approaches, they can account for the temporal aspect of utterances as sequences of words. They do this by retaining a memory of the previous words, i.e., the output of the RNN for the previous word is fed back into the RNN together with the current word. Again, two ways of modelling the data were explored in this research: a set of separate RNNs per schema and a multi-label RNN. The per-schema RNNs allow for assessing the potential added benefit of the deep neural network architecture. For these models, we treat the ordinal scores as separate classes, ignoring the ordering. Each of the nine RNNs in the set outputs a vector of four values between 0 and 1, each value expressing the confidence of the algorithm that the utterance should be assigned a score of 0, 1, 2, or 3 for the specific schema. To obtain the schema score, the score with the highest confidence is selected. The multi-label RNN, on the other hand, can leverage interdependencies between the schemas as it has knowledge of all

schema scores at the same time. It predicts all nine schemas simultaneously and outputs a value between 0 and 1 for each schema. In preparing our analysis script for publication, we encountered the challenge that despite setting all random seeds as required, the trained RNNs showed a small degree of variability in the output when re-running the script. We have therefore chosen a stochastic approach: for both RNN approaches, we first train the models 30 times, we then predict all items of the test set with all 30 models, and finally, we select for the median model in terms of performance. All results reported below are based on the median multi-label RNN and the median per-schema RNN set.

It must be stressed at this point that we only test whether a machine is able to detect patterns at all and do not strive to obtain the best scoring performance. As a consequence, a number of refinement possibilities, such as sequence to vector models or extensive hyperparameter tuning, were not explored.

**H2: Downward arrow converges.**   To examine whether utterances developed with the downward arrow technique converge to a schema, we aimed to predict the algorithm's *scoring accuracy* from the *depth* of utterances. We assigned *depth* = 1 for the automatic thought and increased it incrementally with every downward arrow technique step. Fig 1 shows the number of thought records in our dataset with a specific depth. To determine *scoring accuracy*, we used the predictions made on the test set with the median set of per-schema RNNs of Hypothesis 1. For each utterance, the Spearman correlation between the algorithmically predicted and

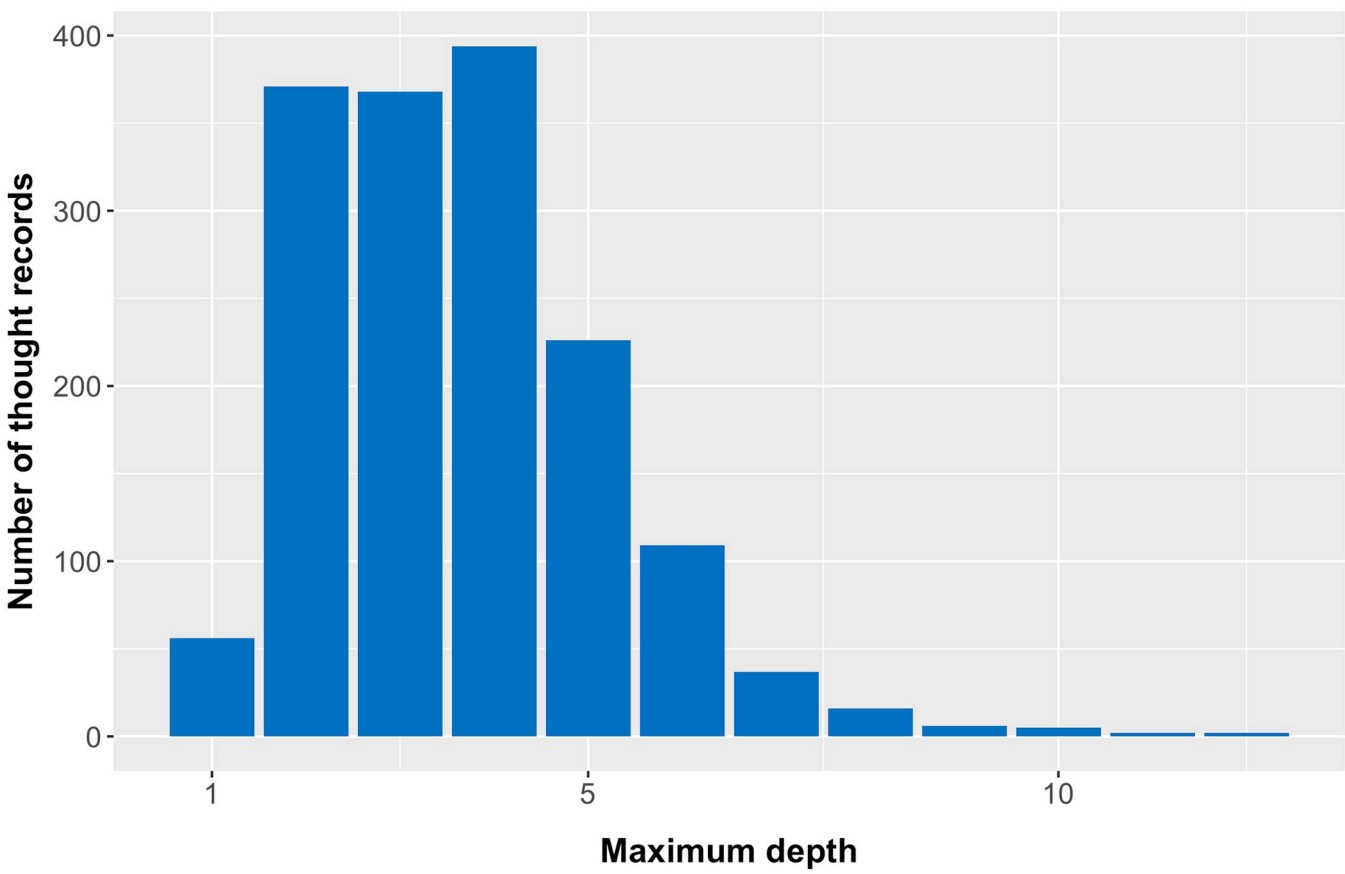

**Fig 1. Distributions of thought records over depth.** Number of thought records having a certain depth, the depth is the number of downward arrow steps + 1 for the automatic thought.

manually assigned scores serves as the measure. Thus, if an utterance such as "I will never be loved" was scored as [3, 0, 0, 0, 0, 0, 0, 1, 0] manually on the nine schemas and received the scores [2, 0, 0, 0, 1, 0, 0, 1, 2] by the RNN, the resulting *scoring accuracy* for this utterance would be $\rho = 0.59$, i.e., the Spearman correlation between the two vectors of scores. To study the effect of depth on scoring accuracy, we conducted a multilevel analysis; the data structure required a three-level linear model with the *depth* as a fixed effect and the *automatic scoring accuracy* as the dependent variable. For each participant (Level 3), there are several thought records (Level 2) and for each thought record, there are several utterances (Level 1). The null model predicts the scoring accuracy from the mean scoring accuracy per participant and thought record. The model therefore has random intercepts at Level 3 and at Level 2 nested within Level 3 (thought records nested in participants). For Model 1, the fixed effect *depth* was added to the null model. We expected to see an increase in automatic scoring accuracy as utterance depth increases.

**H3: Schema patterns are similar across thought record types.** The next analyses tested whether schemas observed in the scenario-based (closed) thought records are predictive of schemas observed in real-life (open) thought records. For this, only the manually assigned scores were used. Each participant completed two achievement-related and two interpersonal closed thought records. The first author labeled all open thought record scenarios as either interpersonal or achievement-related (intercoder agreement with a second independent coder on all open thought records was substantial with Cohen's $\kappa = 0.68$). Nine linear regression models were fit with *schema presence in closed thought records* as the only predictor and *schema presence in the open thought record* as the only outcome variable. Thus, we fit one model for each schema. For example, in the Health schema model, the presence of the Health schema in closed thought records predicts the presence of the Health schema in open thought records. To determine *schema presence in closed thought records*, we identified the two closed thought records with the same situation type (interpersonal or achievement-related) as the open thought record. For each of the nine schemas, we then took the highest score across utterances of both closed thought records and average these two values together. For example, let us assume that a participant described an interpersonal situation in the open thought record. To calculate the predictor for the *Health* schema, the two interpersonal closed thought records of this participant were identified and from each the highest score obtained on the *Health* schema across utterances was taken, leading to two scores, which were then averaged together. We followed the same procedure for the outcome variable, *schema presence in the open thought record*. However, since there is only one such thought record for each participant, no averaging was needed. S2 Appendix illustrates the procedure with a concrete example for clarification.

**H4: Mental illnesses have associated schemas.** The final hypothesis is an exploratory investigation of whether the outcomes from the mental health questionnaires can be predicted from the schema patterns. To this end, we created a summary score per schema and participant. The summary score was calculated by first taking per participant, thought record, and schema the maximum score (0–3) across utterances. This gives one value for each schema for the five thought records a participant completed. These values were then re-coded into a binomial value, with all values smaller or equal to 2 mapping to 0 and 3 mapping to 1. Thus, we only considered schemas that were clearly and unambiguously present. Finally, the binomial values were summed within a participant. Each participant could therefore obtain a maximum value of 5 for a schema if the schema was clearly present in all five completed thought records of the participant. We then created five linear models, each one taking one of the mental health measures (HDAS Depression, HDAS Anxiety, BDI, Cognitive Distortions Relatedness, Cognitive Distortions Achievement) as outcome variable. Every model has the nine schemas as predictors. Since the same data were used to predict five different outcomes, we used a Bonferroni

correction to adjust the significance threshold to $\alpha = 0.05/5 = 0.01$. Just like for Hypothesis 3, the manually assigned scores were used, as this was both suited for testing the hypothesis and less susceptible to errors than the automatically assigned scores.

## Results

To gain insight into the collected data, Table 3 shows the frequencies of each score per schema. In total, there were 5747 utterances.

### H1: Schemas can be automatically extracted

For the majority of schemas, all algorithms could assign scores to the utterances that correlated with the human scores well above what would be expected by chance alone (see Table 4). Furthermore, for all schemas, there was at least one effective algorithm.

As determined with the validation set, the best parameter choice for kNN-C was $k = 4$, while for kNN-R, it was $k = 5$. Both support vector approaches performed best with a radial

**Table 3. Number of utterances with a specific score per schema as manually scored by the first author.** Percentages are provided in parentheses. Schemas are sorted as in the article by Millings & Carnelley [7].

| Schema | Score | | | |
|---|---|---|---|---|
| | 0 (has absolutely nothing to do with schema) | 1 (corresponds a little bit with schema) | 2 (corresponds largely with schema) | 3 (corresponds completely with schema) |
| Attachment | 4047 (70.42%) | 446 (7.76%) | 272 (4.73%) | 982 (17.09%) |
| Competence | 4151 (72.22%) | 314 (5.46%) | 157(2.73%) | 1125(19.58%) |
| Global self-evaluation | 4548 (79.14%) | 226 (3.93%) | 280 (4.87%) | 693 (12.06%) |
| Health | 5428 (94.45%) | 56 (0.97%) | 46 (0.80%) | 217 (3.78%) |
| Power and Control | 5089 (88.55%) | 390 (6.79%) | 154 (2.68%) | 114(1.98%) |
| Meta-cognition | 5626 (97.89%) | 61 (1.06%) | 41 (0.71%) | 19 (0.33%) |
| Other people | 5593 (97.32%) | 92 (1.60%) | 44 (0.31%) | 18 (0.31%) |
| Hopelessness | 4931 (85.80%) | 582 (10.13%) | 174 (3.03%) | 60 (1.04%) |
| Other's views on self | 4688 (81.57%) | 129 (2.24%) | 639 (11.11%) | 29 1(5.06%) |

**Table 4. Spearman correlation and bootstrapped confidence intervals of predicted scores with manually assigned scores per model and schema.** The result of the best model per schema is shown in bold font.

| Schema | Model Outcome | | | | | |
|---|---|---|---|---|---|---|
| | kNN-C | kNN-R | SVM | SVR | per-schema RNNs | multi-label RNN |
| Attachment | 0.55 [0.51,0.60] | 0.63 [0.59,0.65] | 0.65 [0.61,0.68] | 0.68 [0.65,0.70] | **0.73 [0.70,0.76]** | 0.67 [0.66,0.72] |
| Competence | 0.69 [0.64,0.73] | 0.66 [0.63,0.69] | 0.68 [0.65,0.72] | 0.64 [0.61,0.67] | **0.76 [0.72,0.79]** | 0.66 [0.64,0.69] |
| Global self-evaluation | 0.40 [0.33,0.46] | 0.41 [0.36,0.46] | 0.36 [0.31,0.40] | 0.49 [0.45,0.52] | **0.58 [0.54,0.63]** | 0.49 [0.45,0.53] |
| Health | 0.74 [0.65,0.81] | 0.53 [0.44,0.60] | 0.73 [0.65,0.81] | 0.35 [0.31,0.40] | **0.75 [0.65,0.82]** | 0.35 [0.31,0.39] |
| Power and Control | 0.11 [0.02,0.18] | 0.23 [0.17,0.27] | nan [0.00,1.00] | 0.31 [0.26,0.35] | 0.28 [0.20,0.35] | **0.31 [0.27,0.34]** |
| Meta-cognition | nan [0.00,1.00] | 0.10 [0.01,0.20] | nan [0.00,1.00] | 0.11 [0.06,0.16] | -0.01 [0.00,-0.01] | **0.11 [0.06,0.14]** |
| Other people | 0.28 [0.00,1.00] | 0.24 [0.17,0.31] | nan [0.00,1.00] | 0.19 [0.14,0.24] | **0.22 [0.07,0.33]** | 0.16 [0.10,0.20] |
| Hopelessness | 0.48 [0.44,0.55] | 0.51 [0.47,0.56] | 0.49 [0.43,0.53] | 0.54 [0.51,0.57] | **0.63 [0.56,0.68]** | 0.53 [0.50,0.56] |
| Other's views on self | 0.45 [0.41,0.51] | 0.46 [0.42,0.50] | 0.48 [0.43,0.53] | 0.52 [0.48,0.55] | **0.58 [0.52,0.63]** | 0.50 [0.47,0.54] |

The abbreviation *nan* that resulted for some schemas and some models stands for *not a number* and is caused by the absence of variance in the prediction (see text for details).

basis function kernel. The best-performing multi-label RNN was trained in batches of 32 utterances and with 100 epochs. It consists of two hidden layers: an embedding layer, performing the GLoVE embeddings, and a bidirectional long short-term memory layer of 100 nodes. It was trained with a dropout probability of 0.1 and categorical cross-entropy loss. The nine nodes of the output layer use a sigmoid activation function. The metric for choosing the best model was the mean absolute error. The individual models, we set up differently, but adopted some of the hyperparameters of the multi-label model (namely, the batch size, the number of LSTM nodes, the dropout rate, and the loss function). For each schema, the individual models have four outputs, one for each of the four possible scores. The activation function of the final layer is a softmax, to express the likelihood with which a certain utterance has each of the scores.

It can be seen from Table 4 that the per-schema RNNs perform best overall. They take the structure of the data most closely into account, both in terms of the utterances (sequential input) and in terms of the scores (one output neuron per score), and were also able to produce the best predictions for most of the schemas. Any possible advantage of exploiting relationships between schemas was not observable in the results, since the multi-label RNN did not clearly outperform all the other models for any one schema. Interestingly, the *Health* schema is consistently better identifiable by the classification algorithms (kNN-C, SVM, and the per-schema RNNs), while the *Power and Control* schema could be better identified by the regression algorithms (kNN-R, SVR, and multi-label RNN). Nan-values for some algorithms and schemas can be explained by the algorithms predicting 0-values for all items of the test set due to not having seen enough non-zero training examples. Similarly, consistently low correlations for certain schemas (*Meta-cognition*, *Power and Control*, and *Other people*) are the result of a combination of few non-zero training examples (compare Table 3) and variations in the words used within those non-zero training examples. The *Health* schema, for example, could be predicted fairly well, despite few non-zero training examples, because the non-zero training examples had similar wording, frequently including words related to *dieting* and *weight-loss* caused by a scenario with this theme.

## H2: Downward arrow converges

The mean correlation between the predicted schema scores and the manually labeled schema scores was found to be 0.75 ($b = 0.75$, $t(220.76) = 46.97$, $p < 0.001$) when the nesting structure of utterances nested within thought records and thought records nested within participants is taken into account via random intercepts. Steps at a deeper level could not be scored better by the best model of H1 than steps at a more shallow level. The scoring accuracy, as measured by the Spearman correlation, did not improve with additional steps of the downward arrow technique ($\chi^2(1) = 1.21$, $p = 0.27$).

## H3: Schema patterns are similar across thought record types

Fig 2 shows the percentage of utterances having a certain schema (manually assigned score $> 0$) for the open and closed thought records in our dataset. It can be seen that, across participants, schemas are similarly distributed in the two TR types: the mean difference over all schemas is 3.69% with the *Other people* schema having the smallest difference (0.02%) and the Competence schema the largest one (9.24%). Some schemas are more present in open TRs (e.g., the *Power and Control* schema) and others in closed ones (e.g., the *Health* or *Competence* schemas).

On the level of the individual, a series of linear regression models tested whether the active schemas in closed thought records could predict the active schemas in the open thought record

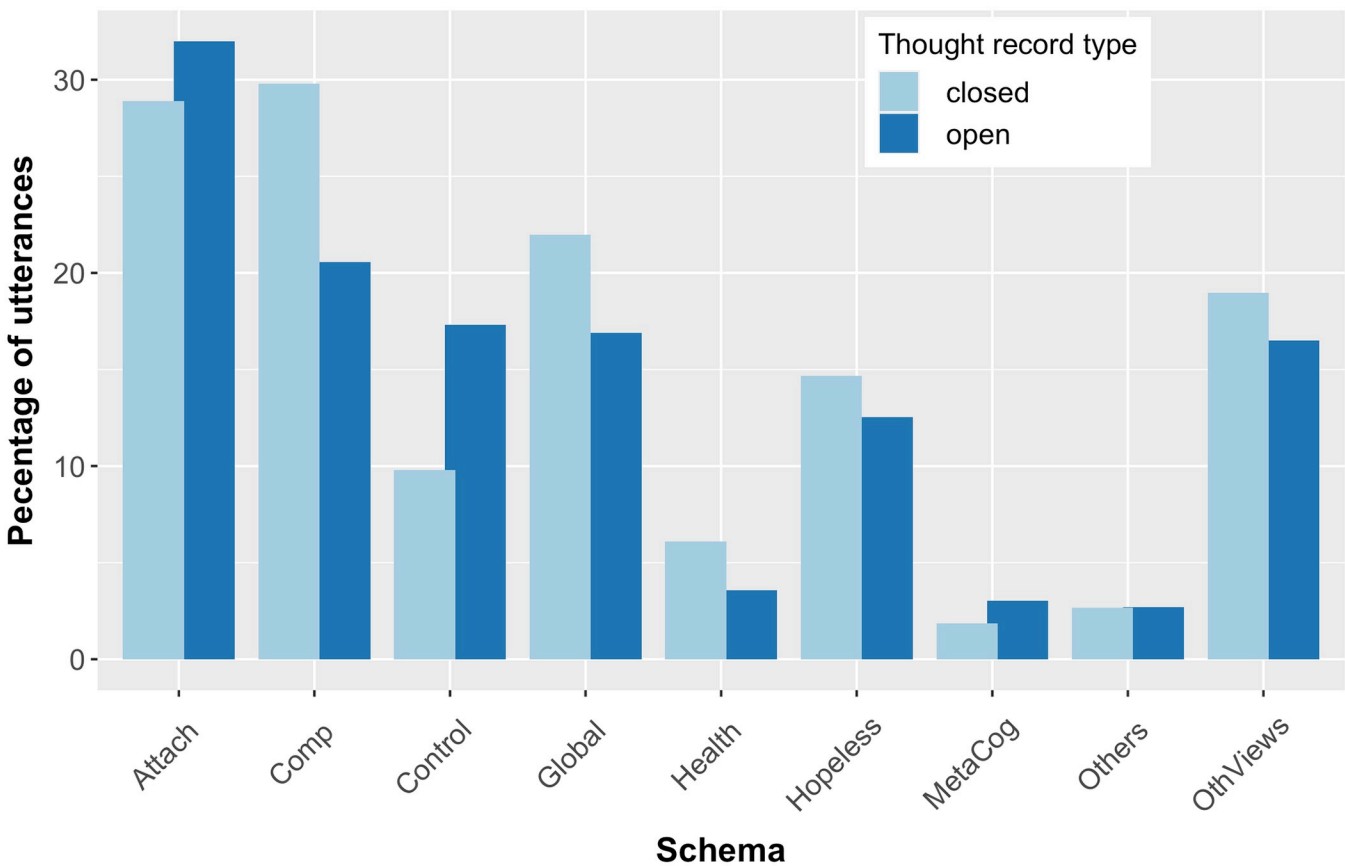

**Fig 2. Presence of schemas in open and closed thought records.** Percentage of utterances that reflect a certain schema (score > 0) in open and closed thought records respectively.

of the same scenario type (interpersonal or achievement-related). The outcome variable was the maximum schema score of the open thought record, while the predictor variable was the average of the maximum schema score of the two closed thought records of the same scenario type. Table 5 presents the results of the models. For the *Competence* schema, 43% of the variance in the open thought record could be predicted from the closed thought records of the same scenario type, while for the *Attachment* schema this was the case for 20% of the variance.

**Table 5. Outcomes of the per-schema linear regression models to test whether participants show similar schema patterns in open as in closed thought records of the same scenario type (interpersonal vs. achievement-related).**

| Schema | b | 95% CI | t | p | F(1,316) | Adj. $R^2$ |
|---|---|---|---|---|---|---|
| Attachment | 0.53 | [0.42,0.65] | 9.07 | <0.001 | 82.28 | 0.20 |
| Competence | 0.79 | [0.69,0.86] | 15.58 | <0.001 | 242.7 | 0.43 |
| Global self-evaluation | 0.31 | [0.18,0.45] | 4.50 | <0.001 | 20.23 | 0.06 |
| Health | 0.21 | [0.07,0.35] | 2.96 | <0.01 | 8.77 | 0.02 |
| Power and Control | -0.08 | [-0.31,0.14] | -0.73 | 0.47 | 0.53 | 0.00 |
| Meta-cognition | 0.17 | [-0.04,0.37] | 1.60 | 0.11 | 2.57 | 0.00 |
| Other people | 0.12 | [-0.03,0.28] | 1.54 | 0.12 | 2.38 | 0.00 |
| Hopelessness | 0.30 | [0.16,0.44] | 4.18 | <0.001 | 17.49 | 0.05 |
| Other's views on self | 0.24 | [0.12,0.37] | 3.77 | <0.001 | 14.22 | 0.04 |

## H4: Mental illnesses have associated schemas

Five linear regression models tested whether there is a link between the active schemas of participants as indicated in thought records and the outcomes on five mental health inventories (see S3 Appendix for table of results). The Bonferroni-corrected $\alpha$ of 0.01 serves as the significance threshold. For both symptom-based mental health inventories for depression, i.e., the HDAS–Depression and the BDI-IA, none of the schemas was a significant predictor of the outcome scores. However, for the anxiety inventory (HDAS–Anxiety) and the two Cognitive Distortion scales, we found that the *Global Self-Evaluation* schema was linked to these measures: all other schemas being equal, any additional thought record with a clearly present *Global Self-Evaluation* schema resulted in a 0.63 ($\beta = 0.18$) point increase on the HDAS–Anxiety ($t = 2.98$, $p = 0.003$), a 2.11 ($\beta = 0.24$) point increase on the Cognitive Distortions—Relatedness measure ($t = 3.83$, $p < 0.001$), and a 2.06 ($\beta = 0.21$) point increase on the Cognitive Distortions—Achievement measure ($t = 3.64$, $p < 0.001$). Finally, the number of thought records with a clearly present *Power and Control* schema also significantly predicted the Cognitive Distortions—Relatedness measure ($b = 3.44$, $\beta = 0.15$, $t = 2.69$, $p = 0.007$).

## Discussion and conclusion

As the first and core hypothesis, we posited that utterances of thought records could be automatically scored with respect to their underlying schemas. With all three machine learning algorithm types (kNN, SVM, and RNN) that were tried, we found affirmative evidence for this. Even when only representing utterances as averages of word vectors, linguistic patterns could be learned (as in the case of the kNN and SVM models). The best-performing algorithm across schemas were the per-schema RNNs. Although for many schemas there is only a small difference in outcome between the best and second-best algorithms, no *one* second-best algorithm emerges. Yet, the fact that the per-schema RNNs outperformed the other algorithms on several schemas provides an indication that the information contained in the word order may be useful for optimal scoring performance. Looking at the best outcomes for each schema, correlations between predicted scores and actual scores ranged from $\rho = 0.11$ to $\rho = 0.81$. The schemas for which the algorithms saw many training examples with non-zero scores (*Attachment* and *Competence*) could be classified well by all. However, the *Health* schema also exhibits good classification potential. This is probably due to very distinctive language as a result of one specific scenario related to dieting and weight loss, i.e., many utterances scored on the *Health* schema contained words such as "fat," "gain," "overweight," "diet," or "skinny." These words are likely to be within close proximity of each other in the word vector space, possibly leading to similar utterance representations and hence a clear linguistic pattern. Although the outcomes from the models cannot be compared directly to the interrater (weighted Cohen's $\kappa = 0.79$) and intrarater (weighted Cohen's $\kappa = 0.83$) reliability scores we obtained on a sample of the data, the reliability scores give a good indication that the nature of the data and the scoring method are limiting the level of agreement that can be achieved as there is some room for interpretation of utterances, schema definitions, and even in the scale points. This, in turn, means that automatic scoring accuracy cannot be expected to exceed the human performance, since the algorithm only has the human-labeled data to learn from. As our goal was only to see whether scoring was feasible and not to obtain the best possible performance, we did not explore many of the other available options for data representation, data augmentation, or modeling. These include looking into more state-of-the-art ways of representing utterances, such as BERT [46] or GPT-3 [47], making better use of the ordering information in the scores, creating a corpus-specific word vector space, or trying to generate more training examples with neural networks. Together with this article, however, we make our collected dataset

publicly available and invite other researchers or machine learning enthusiasts to improve upon our results.

As our second hypothesis, we predicted an algorithm trained on utterances of varying downward arrow technique (DAT) depths to be able to better score the utterances as the depth increases. This is because the DAT was specifically developed to aid patients in identifying their maladaptive schemas, taking the automatic thought from the completed thought record as starting point. After applying the technique, a schema formulation should be reached. In our dataset and with the best performing algorithm of Hypothesis 1, we did not find support for Hypothesis 2. This may be due to only very few participants completing more than three steps. Since our participants were drawn from a non-clinical population and had never practiced thought recording before, it is possible that they did not reach the same level of introspection as a clinical, therapist-guided group would. Additionally, motivations differ between this group of participants (motivated by financial gains) and a clinical group (motivated by mental health gains). Further research might therefore compare our results to those obtained in a clinical setting.

As our third hypothesis, we expected that the dysfunctional schemas that were active when completing scenario-based (closed) thought records would be able to predict those active when completing a real-life personal (open) thought record within participants, given that the closed and open thought records matched in scenario type, that is, both revolved around either an interpersonal or an achievement-related situation. In our study, we relied mostly on pre-scripted scenarios and asked participants to respond to these as if they were real. We found support for our third hypothesis. For two schemas, we even observed that 20% (Attachment schema) and 43% (Competence schema) of the variance in the open thought record score could be predicted by the scores in the closed thought records. This corresponds to the central idea of schema theory: if a person holds a certain schema, this may be activated in various situations of a similar kind and influence how the person appraises the situation [48]. Consequently, we regard it as a viable option to use prescripted scenarios instead of real-life ones when needed. However, it can be argued that the *Attachment* schema may be particularly relevant in *interpersonal* scenarios, while the *Competence* schema plays more in *achievement-related* scenarios. The medium to large (as defined by Cohen [49, p. 413]) effect that shows for these two schemas may therefore be the result of labeling the open thought records as belonging to one of these two scenario types and splitting the dataset accordingly. On the basis of these considerations, the scenarios should be carefully chosen and varied enough to be able to unveil all possible schemas when substituting closed scenarios for open ones. Therefore, a larger number of thought records may be needed than when using open thought records.

Lastly, as our fourth hypothesis, we proposed that the schema patterns across all thought records of a person can predict outcomes on depression, anxiety, and cognitive distortion scales. We found partial support for this hypothesis. Concerning the link between schemas and mental health outcomes, we found no relationship between the schemas and outcomes on both depression inventories. While Millings and Carnelley [7] observed a higher prevalence of the *Power and Control* schema in people with anxious tendencies, we observed higher scores on the HDAS—Anxiety scale when participants had a negative *Global Self-Evaluation*. This schema was also a good predictor of cognitive distortions linked to relatedness and achievement. We could not replicate the finding reported in [7] that higher anxiety scores were linked to a less frequently active *Attachment* schema either. This may, however, be a population effect, as we did not work with a clinical population. Yet, an active *Power and Control* schema was related to more cognitive distortions pertaining to relatedness in our dataset. On the whole, we found more links between schemas and cognitive distortions than schemas and mental health inventory outcomes. This may have to do with thought records being a cognitive task

concerned with unveiling dysfunctional cognitions, which connects directly to the cognitive distortion measure and less to the symptom-based nature of the mental health inventories. Additionally, a single thought record presents a snapshot of a person's thought processes at best and typically many are completed in the course of therapy before a certain schema emerges as clinically relevant [50]. Thus, more extensive experimentation looking for recurring schemas and thought patterns in a clinical population over an extended period of time may paint a clearer picture with regard to the usefulness of the automatic schema-labeling method for therapeutic or diagnostic purposes. On a more practical note, our results indicate that a software application striving to construct a long-term user model might benefit from assigning a higher a priori probability to the activation of the *Global Self-Evaluation* schema after an initial assessment of the user's anxiety levels and cognitive distortions. Still, making a choice on this requires trading off the collection of such sensitive mental health scale data against the added benefit of improving the prediction model. One limitation that must also be considered here is the fact that our results are based on a particular method for combining the utterance labels across all thought records of a participant. This method was a choice and various other methods are conceivable, potentially leading to other outcomes.

The core finding of this research is that it is possible to interpret rich natural language data from the psychotherapy domain using a computer algorithm. The applicability of this finding extends especially to various kinds of psychological assessment. For example, one of the common applications of e-mental health in research are ecological momentary assessments. To date, these typically employ multiple choice response items for self-report measures, which may be combined with sensor readings from handheld devices or wearables (compare [51] for depression). Our findings are promising for effectively using more open response formats and journaling, thus allowing participants to better describe their thoughts, feelings, and behaviors in their own words while minimizing analysis effort. This is also interesting in light of new methodological developments in mental health assessment as a result of big data, such as studying symptom dynamics of individuals with network analyses [52]. Such dynamic networks of symptoms may be augmented with the schemas as determined from thought records to better understand how the activation and co-activation of schemas and other symptoms predicts mental well-being over time. Another possible area of application are cognitive case conceptualizations [53]. These are comprehensive outlines of the patient's problems as first drafted during the intake conversation between patient and therapist. They are continually refined throughout therapy, often on the basis of homework assignments [54]. With the possibility of automatically interpreting thought record data, it may be possible to sketch a first CCC before therapy by collecting and analyzing thought records over the period of time the patient spends on a waiting list and to then collaboratively update this CCC with the therapist as new thought records are completed during therapy. Moreover, Schema Therapy [48] presents a further thought classification system to that of schemas, namely that of schema modes. Furthermore, it proposes a much larger set of schemas than the ones used in this research. With a background in Schema Therapy, it may be possible to use our collected dataset and re-label the data with respect to these other schemas or the schema modes. Beyond psychological assessment, Millings and Carnelley [7] propose future work to compare the derivation of schemas using the downward arrow technique in an online setting to a face-to-face therapy setting. We would be interested in adding the algorithmically derived schemas to this comparison in a long-term study.

In conclusion, we have presented an algorithmic benchmark solution for automatically scoring utterances extracted from thought records with respect to the underlying schema. We expect the model and the opportunities resulting from the positive results to be of relevance for the field of clinical psychology. For the field of computer science, we make the dataset of

collected thought records publicly available. Especially the complexity of the outcome variables (ordinal multi-label) may be intriguing for those looking to develop new algorithms or test existing ones. Lastly, for both fields, clinical psychology and computer science, the dataset could be used to study and advance automatically generated explanations of the algorithmic schema identification. In so doing, it can contribute to diagnoses and explainable artificial intelligence (XAI) technology, which is seen as an important requirement for responsible and effective AI-implementation (e.g., [55]).

## Supporting information

**S1 Appendix. Experimental flow.** Figure displays the different stages of the experiment as traversed by the participants.
(PNG)

**S2 Appendix. Computation of predictor and outcome variables for H3.** Graphical illustration of how we determined the predictor and outcome variables for the nine models of hypothesis 3.
(PNG)

**S3 Appendix. Results of all five linear models to test H4.** Table that summarizes the main outcomes of the five linear models that were fit to assess whether there is a link between schemas and outcomes on various mental health questionnaires.
(PDF)

## Acknowledgments

We would like to acknowledge the help that we received from the two coders who double coded parts of the dataset.

## Author Contributions

**Conceptualization:** Franziska Burger, Willem-Paul Brinkman.

**Data curation:** Franziska Burger.

**Formal analysis:** Franziska Burger, Willem-Paul Brinkman.

**Methodology:** Franziska Burger, Willem-Paul Brinkman.

**Project administration:** Franziska Burger, Willem-Paul Brinkman.

**Software:** Franziska Burger.

**Supervision:** Mark A. Neerincx, Willem-Paul Brinkman.

**Validation:** Franziska Burger.

**Visualization:** Franziska Burger.

**Writing – original draft:** Franziska Burger, Willem-Paul Brinkman.

**Writing – review & editing:** Franziska Burger, Mark A. Neerincx, Willem-Paul Brinkman.

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
