## [Decision Letter · Decision Letter 0]

2 Jun 2021

PONE-D-21-09451

Natural language processing for cognitive therapy: extracting schemas from thought records

PLOS ONE

Dear Dr. Burger,

Thank you for submitting your manuscript to PLOS ONE. After careful consideration, we feel that it has merit but does not fully meet PLOS ONE’s publication criteria as it currently stands. Therefore, we invite you to submit a revised version of the manuscript that addresses the points raised during the review process.

We look forward to receiving your revised manuscript.

Kind regards,

Wajid Mumtaz

Academic Editor

PLOS ONE

Journal Requirements:

Reviewers' comments:

Reviewer's Responses to Questions

**Comments to the Author**

1. Is the manuscript technically sound, and do the data support the conclusions?

Reviewer #1: Yes

Reviewer #2: Yes

2. Has the statistical analysis been performed appropriately and rigorously? 

Reviewer #1: Yes

Reviewer #2: Yes

3. Have the authors made all data underlying the findings in their manuscript fully available?

Reviewer #1: Yes

Reviewer #2: Yes

4. Is the manuscript presented in an intelligible fashion and written in standard English?

Reviewer #1: Yes

Reviewer #2: Yes

5. Review Comments to the Author

Reviewer #1: Paper:

Natural language processing for cognitive therapy: extracting schemas from thought records

Review:

The paper demonstrates an NLP based psychotherapy approach that attempts to automatically detect underlying maladaptive schemas from the thought record forms. For demonstrating that such approach would work, the paper proposes 4 hypothesis H1, H2, H3 and H4 and conducts experiments towards each.

Strength of paper:

The paper explains the methods clearly and contributes a dataset of completed thought records that was collected through the online crowdsourcing platform. Paper discusses the results that supports H1 and partially for H3, H4.

Possible improvements:

Line [265]

[The post-questionnaire comprised three items asking participants how difficult and how enjoyable they found it to complete a thought record, and to indicate how many thought records they think they would complete if they were asked to complete a thought record daily for a period of seven days. ]

How would you ensure that participants are not influenced by secondary motivation of “earning more money” rather than truthfully answering as they are not actual patients but are paid for recording the thoughts. In short, how did you nullify or minimize the inherent monetary motivation for completing the tasks that these participants have but which actual patients won’t.

Table 4

Explain why you get nan as value for Spearman correlation. For Meta-cognition schema, the best correlation is 0.11 which is pretty weak. Explanation of why possibly you got consistently weak correlation across models for this schema, Power and Control and Other People schema.

Abstract mentions that a Pretrained natural language processing software was used. It would be better to clarify whether this nlp software was deep learning based and whether it is pretrained on common text corpus or corpus specific to psychotherapy.

Reviewer #2: I'd like to thank the editors for the opportunity to review this interesting and well written manuscript. So far as I can tell, the study is well done and makes a significant contribution to the emerging literature on the application of artificial intelligence to mental health practice. I have a number of suggested corrections.

In the abstract, the phrase "(Cohen's = 0:79)" should include the kappa that appears elsewhere in the MS and the score should be 0.79.

In Table 1, it's not clear if there were three possible open text entries following the automatic thought entry, or if it is open ended and you simply provided examples of three entries. Figure 1 suggests that it was open-ended, but it would be good to put it in the text.

In the "Materials" section the open vs closed distinction, which is important to your methodology, could be explained in a slightly better organized manner, perhaps by devoting a brief paragraph to each, italicizing the terms or even a bullet point to each paragraph. In each case the term open or closed should come before the definition. My guess is that some readers will refer back to this, and so it's a good idea to make it as clear as possible.

I'd be curious as to what the exact MTurk reimbursement was. I'm not seeing it anyplace.

In Table 2, it's not clear if the example utterances were taken from the automatic thoughts section in Table 1 or from one of the follow-ups.

In the "Schemas can be automatically extracted" section on page 11-12 the descriptions of of each method as they stand will be pretty opaque to anyone who is not a computer scientist. It would be better to devote a paragraph to each, ideally with a description that researchers outside of computer science could follow.

Under H4 on pages 13-14, it's not clear why you dichotomized the schema scores. Dichotomizing can lead to a variety of statistical issues:

https://www.ncbi.nlm.nih.gov/pmc/articles/PMC1458573/

It also might be useful to provide a table including the results of this analysis as you do with H1 and H3.

Thanks for using the Bonferroni correction for H4, and thanks for Table 3. Is there a reason why you included the author's scoring and not the machine scoring? The latter would seem to be most relevant.

In your report on H1 you comment that the per-schema RNNs performed best overall. That is true, but in most cases it's not a dramatic difference between the best and the second best algorithm.

The sentence in your discussion that reads, "Although the outcomes from the models cannot be compared directly to the interrater (weighted Cohen’s κ = 0.79) and intrarater (weighted Cohen’s κ = 0.83) reliability scores we obtained on a sample of the data, the latter give a good indication that close-to perfect automatic is not obtainable," is unclear to me. I think you're saying that the correlations between the algorithm and human classification given in Table 4 are low enough so that the algorithms are likely to be of limited clinical use. I'd agree, but I'm not sure if that's what you're saying. What is close-to-perfect? It's a clinical commonplace that instruments that are used in clinical practice need to be more reliable than those used in research, since clinical practice occurs with individuals rather than ensembles. Is this what you're saying?

Later on you describe the r squared values of .20 and .43 as large. "Comparatively large" would be better.

It would be nice to see some more consideration of implications for mental health practice. However, if you consider that to be outside of your expertise (I'm assuming that you are computer scientists) don't worry about it.

6. PLOS authors have the option to publish the peer review history of their article (what does this mean?). If published, this will include your full peer review and any attached files.

Reviewer #1: No

Reviewer #2: No

---

## [Author Response · Author response to Decision Letter 0]

21 Jul 2021

Dear Editor,

We would first like to express our gratitude for the efforts of the reviewers in helping us improve our manuscript. The remarks have been overall very helpful in pinpointing weaknesses and points of unclarity in the submission. We have addressed each reviewer comment to the best of our ability given the requirements of the journal. Below, we provide the details on how each specific comment was addressed. We further deliver the edited manuscript including tracked changes and line enumeration to facilitate finding back the mentioned changes. 

Reviewer #1

Suggested improvements:

1. Line [265]

[The post-questionnaire comprised three items asking participants how difficult and how enjoyable they found it to complete a thought record, and to indicate how many thought records they think they would complete if they were asked to complete a thought record daily for a period of seven days. ]

How would you ensure that participants are not influenced by secondary motivation of “earning more money” rather than truthfully answering as they are not actual patients but are paid for recording the thoughts. In short, how did you nullify or minimize the inherent monetary motivation for completing the tasks that these participants have but which actual patients won’t.

This is an interesting point raised by the reviewer and one that we had given much consideration in the planning of our research.

As far as the thought recording task itself is concerned, we do not know how a patient population would have responded to the questions and if this had differed. We have added this as an idea for further research to the discussion. We do believe that the monetary motivation should not be nullified but should rather aim to simulate the inherent patient motivation of “wishing to get better.” We had therefore aimed for a fair compensation of participants’ time but no amount that could be seen as very lucrative ($4 for an estimated 35-40 minutes of work). Additionally, the task asked participants to complete many open-text questions. In this, the monetary reward was constant regardless of the effort participants put into the task [we have added lines 286-294 to the manuscript]. Nonetheless, we have emphasized it more in the limitations that further research is required to draw conclusions about patient populations, since we conducted our experiments with a non-clinical population [see lines 677-679].

As far as specifically this post-questionnaire is concerned, we would like to point out, however, that this data was only collected, but not used in the analyses for this manuscript. We collected this data to potentially use in follow-up research but reported it here to give a complete overview here of the methodology and to minimize discrepancies with our OSF-preregistration [compare lines 275-277]. 

2. Table 4

Explain why you get nan as value for Spearman correlation. For Meta-cognition schema, the best correlation is 0.11 which is pretty weak. Explanation of why possibly you got consistently weak correlation across models for this schema, Power and Control and Other People schema.

Nan values can be explained by the fact that the algorithms predict a specific value for all items (in this case 0), leading to no variation in the prediction and hence no possibility of calculating a correlation. This is due to the very low amount of training samples with values higher than 0 for these schemas. Similarly, low numbers of training samples in combination with large variations in wording of thoughts lead to poor predictability in the case of some schemas (Meta-cognition, Power and Control, and Other People). We explain this now in the manuscript [see lines 579-588].

3. Abstract mentions that a Pretrained natural language processing software was used. It would be better to clarify whether this nlp software was deep learning based and whether it is pretrained on common text corpus or corpus specific to psychotherapy.

These were indeed models pretrained on common texts (Wikipedia articles from 2014). Although we had specified it in the manuscript, we agree with the reviewer that this information should already appear in the abstract. We have done this now [see Abstract].

Reviewer #2

1. In the abstract, the phrase "(Cohen's = 0:79)" should include the kappa that appears elsewhere in the MS and the score should be 0.79.

This error was introduced by the automatic pdf-creation service of the PLOS One Editorial Manager. We are grateful to the reviewer for spotting this! We have corrected this now.

2. In Table 1, it's not clear if there were three possible open text entries following the automatic thought entry, or if it is open ended and you simply provided examples of three entries. Figure 1 suggests that it was open-ended, but it would be good to put it in the text.

We thank the reviewer for pointing this out! Each step of the downward arrow technique was in fact a separate open text entry field and participants were asked after each step, whether they wanted to continue with the technique or not. If they wanted to continue, the next downward arrow question would appear. If not, the downward arrow technique stopped, and they were asked to describe the behavior. In elaborating on this in the table, we also noticed that we had forgotten to include the behavior question in the table. We have added this as the final row now [Table 1, caption and table notes, as well as final row].

3. In the "Materials" section the open vs closed distinction, which is important to your methodology, could be explained in a slightly better organized manner, perhaps by devoting a brief paragraph to each, italicizing the terms or even a bullet point to each paragraph. In each case the term open or closed should come before the definition. My guess is that some readers will refer back to this, and so it's a good idea to make it as clear as possible.

Upon reading this section again, we must agree with the reviewer. We have added a few additional sentences to the beginning of the relevant paragraph to better explain this distinction [compare lines 236-241].

4. I'd be curious as to what the exact MTurk reimbursement was. I'm not seeing it anyplace.

The exact MTurk reimbursement was $4 for an estimated 35 minutes of time. We aimed for a fair compensation of participants’ time but did not wish to motivate extra effort with the reimbursement. All participants received the same amount [we have added lines 286-292 to the manuscript].

5. In Table 2, it's not clear if the example utterances were taken from the automatic thoughts section in Table 1 or from one of the follow-ups.

Good point! We have added an additional sentence to the table caption to clarify that the utterances in this table could be either/or [see Table 2, caption].

6. In the "Schemas can be automatically extracted" section on page 11-12 the descriptions of of each method as they stand will be pretty opaque to anyone who is not a computer scientist. It would be better to devote a paragraph to each, ideally with a description that researchers outside of computer science could follow.

We understand the reviewer’s wish to be able to better understand the methods used in this research, particularly when not reading the manuscript as a data scientist. Since we believe that many readers may share this wish, we now describe the methods in some more detail. However, the methods remain technical and not particularly intuitive to those outside of the field. As more detail and a discussion of the methods would be beyond the scope of this paper, we also refer interested readers to textbooks for machine learning [changes to manuscript can be found in lines 402-463].

7. Under H4 on pages 13-14, it's not clear why you dichotomized the schema scores. Dichotomizing can lead to a variety of statistical issues:

https://www.ncbi.nlm.nih.gov/pmc/articles/PMC1458573/

We would like to thank the reviewer for raising this point, but would also like to argue that generally, where summary scores are involved, information is lost and certain statistical issues may arise. Our rationale for choosing this particular dichotomization method was to maximize the chances of a thought record truly reflecting a schema, i.e., only when the schema was clearly present in at least one utterance (schema score of 3) in the thought record, did we count it. We have, however, added it as a limitation in the discussion that we have chosen one particular method to obtain the summary score and that a multitude of other methods is possible, possibly leading to different results [compare lines 725-729]

8. It also might be useful to provide a table including the results of this analysis as you do with H1 and H3.

Although we agree with the reviewer in principle, we had opted not to include such a table in the manuscript originally as there are few significant results or clear patterns emerging from H4, so that a large table would take up a disproportionate amount of space. Instead, we have now opted for including it as an additional appendix.

9. Thanks for using the Bonferroni correction for H4, and thanks for Table 3. Is there a reason why you included the author's scoring and not the machine scoring? The latter would seem to be most relevant.

For both H3 and H4, are about the relationship between two constructs, and not about how well machine learning could link these two constructs. Therefore, the original, manually labeled data was taken, as it does not include errors introduced by the machine learning model [compare lines 546-548].

10. In your report on H1 you comment that the per-schema RNNs performed best overall. That is true, but in most cases it's not a dramatic difference between the best and the second best algorithm.

We have added the consideration raised to the discussion. Although the difference is not dramatic, per-schema RNNs are consistently best, while any of the other algorithms is second-best for at least one schema [see lines 635-638].

11. The sentence in your discussion that reads, "Although the outcomes from the models cannot be compared directly to the interrater (weighted Cohen’s κ = 0.79) and intrarater (weighted Cohen’s κ = 0.83) reliability scores we obtained on a sample of the data, the latter give a good indication that close-to perfect automatic is not obtainable," is unclear to me. I think you're saying that the correlations between the algorithm and human classification given in Table 4 are low enough so that the algorithms are likely to be of limited clinical use. I'd agree, but I'm not sure if that's what you're saying. What is close-to-perfect? It's a clinical commonplace that instruments that are used in clinical practice need to be more reliable than those used in research, since clinical practice occurs with individuals rather than ensembles. Is this what you're saying?

We are grateful to the reviewer for pointing out that this sentence is unclear. It was indeed not our intention to allude to the use of our algorithm in practice at all, but rather we wanted to argue that, due to the nature of the data, some variations exist between human raters in scoring the data. This means that there is a practical limit to the agreement between model and human trainer that can be achieved, as perfect scoring between humans is not feasible for this kind of data and this scoring method. As a result, we cannot expect the algorithm to perform any better than the training data that we supply it with, i.e. the human coding. We have attempted to clarify our argumentation [compare lines 653-658].

12. Later on you describe the r squared values of .20 and .43 as large. "Comparatively large" would be better.

We thank the reviewer for this comment. We had loosely based our qualification of the effects as large on Cohen’s proposed cutoff values for small, medium and large r-squared (Cohen J. (1988). Statistical Power Analysis for the Behavioral Sciences). We have now done so more rigorously (stating that the effects were medium to large) and referring to Cohen for this classification. [change to manuscript in line 696].

13. It would be nice to see some more consideration of implications for mental health practice. However, if you consider that to be outside of your expertise (I'm assuming that you are computer scientists) don't worry about it.

After some discussion, we have decided that this does indeed exceed our knowledge of the mental health domain.

---

## [Decision Letter · Decision Letter 1]

16 Aug 2021

PONE-D-21-09451R1

Natural language processing for cognitive therapy: extracting schemas from thought records

PLOS ONE

Dear Dr. Burger,

Thank you for submitting your manuscript to PLOS ONE. After careful consideration, we feel that it has merit but does not fully meet PLOS ONE’s publication criteria as it currently stands. Therefore, we invite you to submit a revised version of the manuscript that addresses the points raised during the review process.

We look forward to receiving your revised manuscript.

Kind regards,

Wajid Mumtaz

Academic Editor

PLOS ONE

Journal Requirements:

Additional Editor Comments (if provided):

Reviewers' comments:

Reviewer's Responses to Questions

**Comments to the Author**

1. If the authors have adequately addressed your comments raised in a previous round of review and you feel that this manuscript is now acceptable for publication, you may indicate that here to bypass the “Comments to the Author” section, enter your conflict of interest statement in the “Confidential to Editor” section, and submit your "Accept" recommendation.

Reviewer #1: All comments have been addressed

Reviewer #2: All comments have been addressed

2. Is the manuscript technically sound, and do the data support the conclusions?

Reviewer #1: Yes

Reviewer #2: Yes

3. Has the statistical analysis been performed appropriately and rigorously? 

Reviewer #1: Yes

Reviewer #2: Yes

4. Have the authors made all data underlying the findings in their manuscript fully available?

Reviewer #1: Yes

Reviewer #2: Yes

5. Is the manuscript presented in an intelligible fashion and written in standard English?

Reviewer #1: Yes

Reviewer #2: Yes

6. Review Comments to the Author

Reviewer #1: 1. The concern about a possible monetary motivation of participant rather that a real patient's motivation of to get better is satisfactorily addressed as the authors shared relevant details of data collection process.

2. Thank you for explicitly stating the reasons for poor predictability in the case of schemas like Meta-cognition.

3. Thank you for mentioning the corpus of pretrained models. It makes it easier to reproduce or extend this research.

Reviewer #2: I didn't see any major issues. A few minor ones:

In Table 4 it ought to be made clear what "nan" is and the reasons for it. My assumption is nonconvergence, but the authors should discuss it.

Does the fact that the authors' schema for the most part fail to predict depression and anxiety outcomes challenge the validity of the methodology? After all, clinicians primarily care about schema because of their place in the theory underlying cognitive behavioral therapy. I do think that the authors should be more thoughtful in their discussion of this finding.

7. PLOS authors have the option to publish the peer review history of their article (what does this mean?). If published, this will include your full peer review and any attached files.

Reviewer #1: No

Reviewer #2: No

---

## [Author Response · Author response to Decision Letter 1]

6 Sep 2021

Dear Editor,

Thank you for forwarding the second round of reviews to us. We would also like to thank the reviewers for their second evaluation of the manuscript and for pointing out some remaining points-of-improvement.

We have addressed the two points raised by Reviewer #2. Please find below a brief description of how we done this.

We have also submitted the edited manuscript with and without the new tracked changes. 

Reviewer #1 - No additional points raised

Reviewer #2

1. In Table 4 it ought to be made clear what "nan" is and the reasons for it. My assumption is nonconvergence, but the authors should discuss it.

This point had previously also been raised by the first reviewer, and we had addressed it in the text. However, we regard the fact that Reviewer #2 raised it again as an indication that it is still not sufficiently clear. In the table notes, we have now included a short comment stating the meaning of the abbreviation, explaining why the algorithm returns nan-values, and referring to the text for more details [see table notes under Table 4 and lines 565-573 for the in-text details, with the latter added after the previous reviewing round].

2. Does the fact that the authors' schema for the most part fail to predict depression and anxiety outcomes challenge the validity of the methodology? After all, clinicians primarily care about schema because of their place in the theory underlying cognitive behavioral therapy. I do think that the authors should be more thoughtful in their discussion of this finding.

This is an interesting point raised by the reviewer. As the reviewer already figured correctly in the previous reviewing round, we are no psychologists. Our primary interest in testing this hypothesis was not of clinical nature but rather to be able to possibly improve the algorithm. Nonetheless, we do not think that the fact that no link was found challenges or invalidates the methodology. This is because the methodology is based on thought records, which are momentary assessments of a person’s thoughts, while the way in which depression and anxiety outcomes are assessed is with inventories that are symptom-based and longer term. However, if the methodology is valid (and we believe it to be), we would certainly expect to see such links showing in a clinical population that regularly completes thought records over an extended period (as is also done in therapy) and holds schemas which the algorithm can predict relatively reliably (e.g., Attachment or Competence schemas). We have added a few sentences to express this in the discussion [see lines 704-710].

---

## [Decision Letter · Decision Letter 2]

13 Sep 2021

Natural language processing for cognitive therapy: extracting schemas from thought records

PONE-D-21-09451R2

Dear Dr. Burger,

We’re pleased to inform you that your manuscript has been judged scientifically suitable for publication and will be formally accepted for publication once it meets all outstanding technical requirements.

Kind regards,

Wajid Mumtaz

Academic Editor

PLOS ONE

Additional Editor Comments (optional):

Reviewers' comments:

Reviewer's Responses to Questions

**Comments to the Author**

1. If the authors have adequately addressed your comments raised in a previous round of review and you feel that this manuscript is now acceptable for publication, you may indicate that here to bypass the “Comments to the Author” section, enter your conflict of interest statement in the “Confidential to Editor” section, and submit your "Accept" recommendation.

Reviewer #2: All comments have been addressed

2. Is the manuscript technically sound, and do the data support the conclusions?

Reviewer #2: Yes

3. Has the statistical analysis been performed appropriately and rigorously? 

Reviewer #2: Yes

4. Have the authors made all data underlying the findings in their manuscript fully available?

Reviewer #2: No

5. Is the manuscript presented in an intelligible fashion and written in standard English?

Reviewer #2: Yes

6. Review Comments to the Author

Reviewer #2: (No Response)

7. PLOS authors have the option to publish the peer review history of their article (what does this mean?). If published, this will include your full peer review and any attached files.

Reviewer #2: No

---

## [Editor Report · Acceptance letter]

8 Oct 2021

PONE-D-21-09451R2 

Natural language processing for cognitive therapy: extracting schemas from thought records 

Dear Dr. Burger:

I'm pleased to inform you that your manuscript has been deemed suitable for publication in PLOS ONE. Congratulations! Your manuscript is now with our production department. 

Kind regards, 

on behalf of

Dr. Wajid Mumtaz 

Academic Editor

PLOS ONE